# Unveiling the Potentiality of Shikonin Derivatives Inhibiting SARS-CoV-2 Main Protease by Molecular Dynamic Simulation Studies

**DOI:** 10.3390/ijms24043100

**Published:** 2023-02-04

**Authors:** Raju Das, Sarmin Ummey Habiba, Raju Dash, Yohan Seo, Joohan Woo

**Affiliations:** 1Department of Physiology, Dongguk University College of Medicine, Gyeongju 38067, Republic of Korea; 2Department of Pharmacy, BGC Trust University Bangladesh, Chittagong 4381, Bangladesh; 3Department of New Biology, Daegu Gyeongbuk Institute of Science and Technology, Daegu 42988, Republic of Korea; 4New Drug Development Center, Daegu Gyeongbuk Medical Innovation Foundation, Daegu 41061, Republic of Korea; 5Channelopathy Research Center (CRC), Dongguk University College of Medicine, 32 Dongguk-ro, Goyang 10326, Republic of Korea

**Keywords:** SARS-CoV-2, main protease, shikonin derivatives, molecular docking, molecular dynamics simulation

## Abstract

Shikonin, a phytochemical present in the roots of *Lithospermum erythrorhizon,* is well-known for its broad-spectrum activity against cancer, oxidative stress, inflammation, viruses, and anti-COVID-19 agents. A recent report based on a crystallographic study revealed a distinct conformation of shikonin binding to the SARS-CoV-2 main protease (M^pro^), suggesting the possibility of designing potential inhibitors based on shikonin derivatives. The present study aimed to identify potential shikonin derivatives targeting the M^pro^ of COVID-19 by using molecular docking and molecular dynamics simulations. A total of 20 shikonin derivatives were screened, of which few derivatives showed higher binding affinity than shikonin. Following the MM-GBSA binding energy calculations using the docked structures, four derivatives were retained with the highest binding energy and subjected to molecular dynamics simulation. Molecular dynamics simulation studies suggested that alpha-methyl-n-butyl shikonin, beta-hydroxyisovaleryl shikonin, and lithospermidin-B interacted with two conserved residues, His41 and Cys145, through multiple bonding in the catalytic sites. This suggests that these residues may effectively suppress SARS-CoV-2 progression by inhibiting M^pro^. Taken together, the present in silico study concluded that shikonin derivatives may play an influential role in M^pro^ inhibition.

## 1. Introduction

A novel coronavirus, named COVID-19 by the World Health Organization, has gained global concern after the rapid outbreak of pneumonia in Wuhan, China [1,2]. Typically, coronaviruses (CoVs) belong to the Coronaviridae family [3]. Three genetically identical human strains (SARS-CoV, SARS-CoV-2, and MERS CoV) are susceptible to infection of the upper and lower respiratory tracts and to infecting underlying cardiac patients [4]. SARS-CoV-2 is characterized as a positive-sense single-stranded RNA virus with an envelope [5,6]. Viral genome assembly of SARS-CoV-2 reveals several structural and non-structural proteins, including spike glycoprotein, membrane protein, envelope protein, nucleocapsid protein, and 16 non-structural proteins (nsp1-16). During infection, structural proteins play a role in new virion assembly [7,8,9,10]. Two open reading frames (ORF1a and ORF1b) encode two protease enzymes, papain-like protease (PL^pro^) and 3C-like protease (3CL^pro^). Additionally, these frames encode RNA-dependent RNA polymerase and endoribonuclease [10,11,12]. Each structural protein is known to serve individual functions: the spike protein recognizes the binding site to the host cell surface, and the E protein is involved in viral morphogenesis [13]. Drug repurposing strategies have already been suggested for several targets [14], although there is currently no significant outcome to combat infection. A number of vaccines have already been established in a short time against SARS-CoV-2 infection [15]. However, due to mutations, it is still necessary to discover more effective vaccines or drugs that can terminate SARS-CoV-2 infection [16].

Many antiviral drugs that target protease enzymes have been marketed and used for HIV/AIDS and hepatitis C viruses. Therefore, the main protease (M^pro^) inhibitor of SARS-CoV-2 can unveil a potential treatment for COVID-19 patients. M^pro^ refers to 3-chymotrypsin-like cysteine protease (3CL^pro^) and mainly interferes with two overlapping polyprotein (ppla and pplab) cleavage processes [17,18,19,20,21], which are cleaved into 16 non-structural proteins that are essential for replication. Thus, blocking the main protease impedes viral replication and transcription. The recent release of several highly resolved crystallographic structures has allowed scientists to gain more insights into ligand-binding modes and to resolve conserved residues that are highly enriched in ligand recognition.

In the quest for a potent inhibitor with high inhibitory affinity and efficacy, numerous ligand-bound M^pro^ complexes have shed light on M^pro^-based drug discovery [22,23,24]; however, the shikonin-M^pro^ crystal complex appears in a distinct form and provides an invaluable resource for designing novel antiviral agents [25]. Shikonin, a natural bioactive compound derived from the roots of *Lithospermum erythrorhizon,* belonging to the Boraginaceae family, has several medicinal properties [26,27,28]. In cancer, shikonin exerts its broad-spectrum effect by inducing apoptosis, autophagy, and the necroptosis pathway in cell proliferation and migration processes [29]. In inflammatory conditions, it reduces proteasomal activity through the inhibition of NF-κB and ubiquitinated proteins, as reported by in vitro and in vivo studies [27]. Shikonin modulates several signaling pathways, including the PI3K, AKT/mTOR, and MAPK pathways. In some cases, shikonin derivatives are more effective than shikonin alone. Shikonin was reported to act as an M^pro^ inhibitor against SARS-CoV-2 with an acceptable IC_50_ value and binding free energy. This suggests the possibility of designing a better inhibitor based on shikonin derivatives [22,30]. Based on this hypothesis, the present study implemented a computational workflow (Figure 1), including a structure-based drug design approach, to identify potent M^pro^ inhibitors using shikonin derivatives based on the shikonin-M^pro^ complex.

## 2. Results

### 2.1. The Overall Architecture of the SARS-CoV-2 Main Protease

The crystal structure of the SARS-CoV-2 M^pro^ is a homodimer, where each protomer is composed of three domains and an extended loop region that connects domains II and III. Domain III initiates the dimerization process by interlinking a salt bridge between two protomers, thus accommodating a central core for ligand binding between domains I and II [31,32]. As a cysteine protease, the structure of M^pro^ provides a catalytic dyad bearing two critical residues, Cys145 and His41, which play an essential role in catalysis [22]. Moreover, M^pro^ contains an N-terminal domain, oxyanion loops [33], and a C-terminal domain (where the N-terminal domain is directly involved in auto-cleavage activity). Oxyanion holes, which are often formed by histidine residues, participate in enzyme catalysis and facilitate enzyme activity. In addition, oxyanion holes belong to the negatively charged residue, Glu166, and the positively charged His41, His163, and His172 residues, found in the active conformation. However, this oxyanion hole is more disorganized in the entire structure. The overall graphical docking workflow is shown in Figure 2.

To facilitate rational drug design and unveil the substrate-binding mechanism, several SARS-CoV-2 M^pro^ co-crystal structures have been elucidated. Most SARS-CoV-2 M^pro^ structures confirmed several covalent and non-covalent interactions during substrate binding, where residues His41 and Cys145 are common in all compounds; thus, showing conservation of these two residues in substrate recognition [34,35]. In shikonin-bound M^pro^, residue His41 located on the S2 subsite interacts with the shikonin aromatic head group, while Cys145 on the S1 subsite interacts with shikonin through hydrogen bonding [25]. Additional hydrogen bond interactions were also observed on the S3 subsite between the hydroxy and methyl groups of shikonin with residues Arg188 and Gln189.

### 2.2. Molecular Docking

To validate our hypothesis of finding an effective lead against M^pro^, molecular docking was performed using Glide XP docking methodology (Figure 3). The accuracy of docking was first verified using a re-docked methodology, in which the native co-crystal ligand was first separated from the complex and re-docked into the same ligand-binding site. The retained docked conformation was compared with that of the original complex, and the root mean square deviation (RMSD) was determined. An RMSD value of 0.07 Å was observed when the docked complex was compared with the native complex, suggesting that the docking method can accurately predict ligand-binding conformation. Following the same protocol, 20 shikonin derivatives were subjected to molecular docking and MM-GBSA prediction; the results are shown in Appendix A. After considering the almost identical binding position compared to crystal shikonin structure, and the highest score for binding free energy, four hit compounds were selected. Among them, shikonin glucoside obtained the highest docking score (−8.079 kcal/mol), while alpha-methyl-n-butyl shikonin showed the lowest docking score (−5.631 kcal/mol). The docking score of control shikonin was also obtained (−6.029 kcal/mol).

### 2.3. Drug-Likeness Prediction Studies

Relevant physiochemical descriptors were considered for the four selected hits to predict drug-likeness properties using the QikPropscript of Schrodinger. The QikProp properties and descriptors cover 95% of all known drugs based on Lipinski’s “rule of five” [36]. After analysis using the rule of five, shikonin glucoside violated one rule in terms of the number of hydrogen bond acceptors (HBA) [37]. The partition coefficient (QPlog Po/w) was within the acceptable range of −2 to 6.5 for all hits. Cardiac toxicity assessment, another important drug-likeness property, is important for controlling cardiac activity via the HERG K^+^ channel [37]. Other than shikonin glucoside and lithospermidin-B, the compounds were within a suitable range (logHERG > −5). To be an effective drug, it must be absorbed at the right time and distributed throughout the body to maximize its metabolism and action. As a result, the human oral absorption (HOA) was calculated, where a value of 1 indicates low absorption and a value of 3 indicates high absorption. Results showed that shikonin glucoside had low absorption, whereas the other compounds had high absorption. The Madin–Darby canine kidney (MDCK) is an additional parameter used to assess permeability. The results of the analysis indicate that beta-hydroxyisovaleryl shikonin and alpha-methyl-n-butyl shikonin will absorb better, although the estimated value of shikonin glucoside is outside the recommended range. The efficacy of hypothetical hits also depends on QPlog Khsa parameters, which estimate binding to human serum albumin. The recommended range is −1.5 to 1.5. Subsequently, the projected hits of selected hits were compliant with this parameter, suggesting that these substances would readily circulate throughout the serum plasma. The overall ADME/T parameters and results are depicted in Appendix A.

### 2.4. Molecular Docking Analysis

Based on detailed XP docking analysis, non-bonded interactions were considered for protein–ligand interaction profiling. A summary of the molecular interactions is provided in Appendix A and Figure 4. According to the interaction analysis, shikonin binds to the M^pro^ substrate-binding core with multiple interactions. The naphthazarin moiety of shikonin formed a π-stack with the interacting residue His41 located in subsites S1 and S2; however, residue Arg188 showed hydrogen interaction with the head hydroxyl group of the naphthazarin ring. Additional hydrophobic interactions were also observed between shikonin and residues Met165, Glu166, and Asp187. Shikonin glucoside was stabilized in the binding domain by forming a double hydrogen bond with the Glu166 and Thr190 residues, and forming a single hydrogen bond with the Gln192 residue. Shikonin glucoside also formed four hydrophobic interactions with residues His41, Met49, Met165, and Gln189, as well as Π-stacking interactions with His41 to occupy the pocket. Another candidate compound, alpha-methyl-n-butyl shikonin, formed numerous hydrophobic interactions with residues His41, Met49, Leu167, Glu189, and Glu192. In addition, the conserved residue, Arg188, forms a double hydrogen bond with the side chain of the candidate compound. Beta-hydroxyisovaleryl shikonin exhibits significant hydrophobic interactions with the conserved residues Met165 and Glu192, and exhibits multiple hydrogen bond interactions with the residues His164, Glu166, and Asp187, among other residues. Lithospermidin-B also has a higher docking score and binding free energy than control shikonin, and its side chain forms a double hydrogen bond with Thr190 and a single hydrogen bond with Glu192. A hydrophobic interaction was observed between the lithospermidin-B naphthazarin moiety and His41. In addition, a second hydrophobic contact was observed between Met49 and Leu16.

Based on shikonin-M^pro^ (PDB:7CA8) and other complex 2D interactions, His41 and Cys145 were the primary conserved residues involved in ligand recognition, and other important residues were also noticeable; however, in the current molecular docking 2D interaction, the lack of an exact role of other crucial residues in substrate binding contributed to more accurate predictions. Thus, dynamic simulation studies are analyzed further in the following section.

### 2.5. Molecular Dynamics Simulations

We performed a total of 700 ns of MD simulations for both the control and selected hits of the M^pro^-ligand complexes. First, the simulated trajectories were originally analyzed for RMSD, which indicates the conformational stability of the protein during the simulation. After completion of all simulations, RMSD analysis showed that all proteins, including the control and hit complexes, reached equilibrium and were stable after 200 ns (Appendix A). As a result, the final 500 ns trajectories of all systems, including control and hits, were considered and concatenated for further investigation.

According to Appendix A, apo remained stable throughout the simulation time without any significant variation; however, an upward trend was evident from 550 to 700 ns. This suggests that the stability of apo was maintained throughout the simulation period. Shikonin induced a large fluctuation starting from 2 Å up until 6 Å at 350 ns; however, it maintained stability after 350 ns to the end of the simulation period. After 220 ns, beta-hydroxyisovaleryl shikonin remained stable until the end of the simulation period, as lithospermidin-B showed several fluctuations throughout the simulation time. Alpha-methyl-n-butyl shikonin showed fluctuations from 200 to 250 ns and 380 to 450 ns, after which it remained stable until the end of the simulation.

According to all the ligand RMSD analyses, apo and all hits remained stable after 600 ns. Therefore, the trajectories of the last 100 ns were considered for MM-GBSA calculations, in which shikonin glucoside showed less energy than the other hit compounds. As a result, apo, shikonin, and other hits were selected for further analysis, with the exception of shikonin glucoside. The results related to the docking score and binding free energy are shown in Appendix A.

#### 2.5.1. Effect of Ligands on Conformational Stability

Quantifying the radius of gyration (Rg) of each trajectory allowed us to examine how the presence of various ligands influences the conformational stability of M^pro^, where lower Rg values correspond to high compactness. As shown in Figure 5, the Rg of all trajectories reflected changes in the overall shape of the protein structure caused by ligand binding. Overall, Rg analysis indicated a significant difference between shikonin and the selected hits, while lithospermidin-B exhibited the most variation in comparison to the other systems. The lithospermidin-B complex had an Rg value of 3.9–4.4 Å, whereas the shikonin complex had an Rg value of 3.4–3.8 Å. In contrast, alpha-methyl-n-butyl and beta-hydroxyisovaleryl shikonin showed that protein flexibility and folding compactness did not change much compared with lithospermidin-B. Overall, the data indicate that alpha-methyl-n-butyl shikonin and beta-hydroxyisovaleryl shikonin maintained stable compactness when compared to the control shikonin. However, lithospermidin-B significantly enhanced flexibility in protein folding during the entire simulation time.

The term “root mean square fluctuation” (RMSF) is used to describe local changes throughout the protein chain; therefore, we performed RMSF analysis using simulation trajectories (Figure 6). The highest peak represents the fluctuating protein regions that are more flexible during the whole simulation periods, where all simulation systems exhibited a protein RMSF of 0.5–11 Å. As seen in Figure 6C, lithospermidin-B induced fluctuations that were significantly noticeable throughout the protein. The highest variation recorded was approximately 6 Å in Domains III; this might be indicative of major ligand interactions. However, alpha-methyl-n-butyl and beta-hydroxyisovaleryl shikonin showed fewer fluctuations than lithospermidin-B. In addition, the RMSF values of alpha-methyl-n-butyl and beta-hydroxyisovaleryl shikonin were similar to those of the apo form, indicating the stability of these two ligands compared to lithospermidin-B.

#### 2.5.2. Conformational Dynamics of Protein–Ligand Binding

The ligand flexibility inside the active site of the receptor was also calculated to better understand the formation and breakdown of new bonds. Shikonin remained stable till 600 ns; afterwards, a fluctuation was recorded around 2 Å and maintained till 700 ns. Alpha-methyl-n-butyl shikonin fluctuated until 200 ns, then decreased and remained stable until the end of the simulation time. Unlike shikonin and alpha-methyl-n-butyl shikonin, beta-hydroxyisovaleryl shikonin and lithospermidin-B showed different conformations throughout the simulation (Appendix A).

To further elucidate ligand selectivity and interaction patterns, the total number of protein–ligand contacts was counted from the total interaction panel during the simulation periods. Thus, the total contact between the protein and ligand, as well as the residual contribution to ligand binding for each complex, were determined and are shown in Figure 7. As shown in Figure 7A, the shikonin-mediated total percentage of protein–ligand contacts was mostly covered by the conserved residue His41 by a significant number of hydrogen and hydrophobic interactions, although the number of other crucial residues reported in the discovered crystal shikonin-M^pro^ structure was hardly seen. On average, only a few hydrogen and water bridge contacts were observed. However, Met165 exhibited a high number of hydrophobic interactions throughout the simulation trajectories. As expected, a robust interaction pattern was observed between alpha-methyl-n-butyl shikonin and beta-hydroxyisovaleryl shikonin. Meanwhile, lithospermidin-B made insignificant contact with the M^pro^ binding site, which may be predicted based on the RMSD and Rg analyses. The strong hydrophobic and polar interactions mediated by alpha-methyl-n-butyl shikonin and beta-hydroxyisovaleryl shikonin with His41, Met49, Cys145, Met165, Glu166, Arg188, Thr190, and Gln 192 are recognized to be essential residues in the contribution of ligand recognition, as presented in Figure 7B,C.

Relevant to the shikonin-M^pro^ crystal structure, control shikonin and the selected top hits perceived the same kind of interaction with crucial residues in the substrate binding site of M^pro^, where residues Glu166, Met165, and Arg188 gained potential interest in ligand recognition in the case of the top three hits; however, the involvement of two conserved residues, His41 and Cys145, in ligand binding was hardly seen except for in lithospermidin-B (Figure 7D). Interestingly, the stability of M^pro^ induced by beta-hydroxyisovaleryl shikonin was significantly mediated by conserved His41 and Cys145, confirming the role of these residues in complex formation. This analysis is also consistent with the results of the RMSD and Rg analyses.

#### 2.5.3. Ligand-Induced Changes on Protein Dynamics

Furthermore, we performed principal component analysis (PCA), which uses eigenvalues and eigenvectors, to check the atom’s most active motion. The value of the covariance matrix was used to define the contribution of each atom. We compared the essential subspaces obtained from PCAs of the control and hit ensembles using the root mean square inner product (RMSIP). The RMSIP values quantify the similarity and dissimilarity of essential subspaces by providing high and low values within a range of 0 to 1. According to the analysis in Figure 8, the pairwise comparison of wild and compound bound structures revealed that compound-induced systems exhibit a significantly different dynamical motion than wild-type structures, which account for the majority of dynamical activities. To illustrate the overall conformational changes based on the two most common principal components (PC1 and PC2), a two-dimensional (2D) plane was projected to describe the resulting conformational landscape, where each dot represents a blue to red structure.

The bottom panel represents Figure 9 showing the PCA results for the wild-type and hit complexes, where the first PCs were identified for apo, shikonin, beta-hydroxyisovaleryl shikonin, alpha-methyl-n-butyl shikonin, and lithospermidin-B. The representative PCs in the wild-type, shikonin, beta-hydroxyisovaryl shikonin, alpha-methyl-n-butyl shikonin, and lithospermidin-B accounted for 53.23, 50.87, 43.73, 37.65, and 46.44% of the total variance in motion, respectively. The overall total variance for apo, shikonin, alpha-methyl-n-butyl shikonin, beta-hydroxyisovaryl shikonin, and lithospermidin-B were 92, 92, 90, 87, and 96%, respectively. As shown in the right panel of Figure 9, all the first PCAs (excluding PC1) exhibited dynamic movement variations. The wide tubes in the illustration indicate atom displacement, whereas the thin tubes indicate simulation rigidity. Compared to the apo and hit complexes, most atomically significant displacements were observed in lithospermidin-B, particularly in domain III rather than in domains I and II. In contrast, alpha-methyl-n-butyl shikonin and beta-hydroxyisovaryl shikonin-induced atomic displacement was not significantly observed in the overall structure. On the other hand, both alpha-methyl-n-butyl shikonin and beta-hydroxyisovaleryl shikonin (blue and red dots, respectively) were dispersed in a scattered state. The majority of the red and blue dots were concentrated in beta-hydroxyisoveryl shikonin and alpha-methyl-n-butyl shikonin, and the majority of conformations were distributed in the center Figure 9A–E. However, when lithospermidin-B was viewed, the blue and red dots were barely visible. As a result, it may be stated that lithospermidin-B was more flexible in motion during the simulation than beta hydroxyisovaleryl shikonin and alpha-methyl-n-butyl shikonin.

Next, we evaluated the dynamic cross-correlation matrix (DCCM), which provided an overall picture of the correlated and anti-correlated protein motions during the simulation [38,39]. The red areas in the correlation maps show a positive value (+1), which represents a strong correlation between the movement of residues i and j. In contrast, the blue regions denote a high degree of anti-correlation between the activities of the residues, characterized by a negative value (−1). According to the correlation map, the apo and hits showed significant differences in both correlated and anti-correlated motions, as shown in Figure 10A–E. Overall, analysis according to Figure 10 indicates that residues 10 to 99 present in domain I showed increased negative correlations with domain II residues (100-182). However, there was no significant positive or negative motion seen in domain III compared to domain I. In addition, domain III residues 202–225 also showed high anti-correlation with domain II. Therefore, the residues of domain II indicate the most moveable region of the whole structure. However, most of the areas between domains I and III did not participate in positive or negative motion throughout the simulation period.

According to Figure 10B–E, binding of shikonin, beta-hydroxyisovaleryl shikonin, and alpha-methyl-n-butyl shikonin initiates a negative correlation in both domains I and III, although it is slightly lost in the lithospermidin-B complex; however, significant positive motion was observed in domain II of all systems. Thus, the overall DCCM analysis suggests that the compounds directly modulate the structural stability.

## 3. Discussion

Globally, the new coronavirus has caused a significant public health disaster due to its widespread distribution. To date, several synthetic and non-synthetic compounds came into discussion targeting M^pro^ of SARS-CoV-2. However, due to the lack of selectivity, lower affinity, and toxicity, these compounds are now out of the limelight. An effective compound requires high affinity, low toxicity, and minimal side effects, all of which are determined by its physical and chemical properties. In this sense, natural molecules have gained priority over synthetic compounds in recent decades as effective antiviral agents.

The latest discovery of the M^pro^ structure of COVID-19 has enabled the identification of active ligand-binding sites and crucial residues, which provide a possible opportunity to identify potential therapeutic candidates. Following that, the shikonin-M^pro^ crystal structure is a new form, in which shikonin showed a new mode of binding in the case of M^pro^ inhibition, unveiling the possibility of designing shikonin derivatives as M^pro^ inhibitors. In this study, a structure-based drug design (SBDD) method was used to screen out potential shikonin derivatives employing algorithm-based molecular docking and molecular dynamics simulations targeting M^pro^ of SARS-CoV-2. Molecular docking provides a feasible protein-ligand binding pose. Data from docking and 2D interactions showed that shikonin and the selected hits interacted with the conserved residue His41, as well as with other crucial residues (Met165, Gln192, and Arg188) [25]. All of these residues are involved in ligand recognition, which is also observed in all M^pro^ structures. In addition, residue His41 in the shikonin-M^pro^ crystal complex formed a Π-stacking interaction on the S2 subsite, which was similarly observed in control shikonin, beta-hydroxyisovaleryl shikonin, and shikonin glucoside. Similarly, Arg188 formed a hydrogen bond with the alpha-methyl-n-butyl shikonin compound in this study. In addition, all M^pro^ structures included Glu166, which is highly conserved and forms a hydrogen bond with beta-hydroxyisovaleryl shikonin and shikonin glucoside.

Molecular dynamics simulations are used to anticipate the behavior of atoms under dynamic biological conditions, such as protein–ligand interactions, ligand stability inside the receptor proteins, and conformational changes of the ligand or macromolecules, all of which occur in biological conditions. MD simulation results showed that shikonin and selected hits equilibrated after a 600 ns simulation period. The estimated energy related to MM-GBSA for the last 100 ns revealed a higher binding free energy in favor of beta-hydroxyisovaleryl shikonin and lithospermidin-B than other hits, in which shikonin glucoside showed less binding free energy compared to all compounds, supporting the reason that the ligand moved away at the end of the simulation period. In all the protein–ligand contacts, residues Met165, Thr190, and Gln192 maintained the stability of alpha-methyl-n-butyl shikonin by forming numerous hydrogen bonds and hydrophobic interactions. This was also observed in the RMSF analysis (Figure 6). Surprisingly, beta-hydroxyisovaleryl shikonin maintained stability through all conserved residues, His41, Cys145, Met165, Glu166, Arg188, and Gln192, as reported in a previous study.

Furthermore, we investigated the ADMET profile of selected bioactive compounds by using Quickprop, where, based on several descriptors, compounds are sorted by #stars ranging from 0–5; a high number indicates low drug-like properties. Results from the ADMET properties indicated that alpha-methyl-n-butyl shikonin has high drug-like properties. Therefore, accumulating data from a large number of studies suggest that the compound alpha-methyl-n-butyl shikonin may have the potential to terminate the replication process of the M^pro^ of SARS-CoV-2, although in vivo and in vitro studies are needed for further validation.

## 4. Materials and Methods

### 4.1. Ligand Preparation

Based on a literature search, 20 compounds were identified and sorted according to their chemical structure, IUPAC name, InChi key, etc., from the PubChem databases [40,41,42,43]. The 2D structures of all compounds were retrieved and subjected to 3D structure generation using the ligand preparation wizard module in Schrödinger Suite, keeping the pH at 7.0 ± 2.0, to generate low-state, right chiralities, and structure-specific ionization conditions of each ligand [44]. The entire methodology was performed under the OPLS3 force field, and 32 stereoisomers were obtained from each ligand. Subsequently, the final resulting compounds were allowed to perform an extra glide precision (XP) module in Schrodinger Suite.

### 4.2. Protein Preparation and Grid Generation

The selected protein structure (PDB:7CA8) for molecular docking was retrieved from the RCSB protein data bank and subjected to the multistep protein preparation wizard of the Schrödinger Suite 2017-1 program to correct the structure [45,46]. Subsequently, the structure was pre-processed by the insertion of relevant H-atoms, charges, and bond orders, as well as the removal of unnecessary water from the active site that extends beyond 5 Å. Afterwards, structure optimization was carried out by using PROKA at neutral pH. To perform final restrained minimization, we used the OPLS3 force field, in which the convergences of heavy atoms were limited to RMSD 0.30 Å. For the docking simulation, the active site of the protein was established by building a grid box around the binding site of the reference ligand. For post-minimization, the grid generation settings were set at their default values with a box size of 15 Å × 15 Å × 15 Å, and the OPLS 2005 force field was used for post-minimization. The charge cut-off and van der Waals scaling factor were set to 0.25 and 1.00, respectively [47].

### 4.3. Molecular Docking Simulation

We used the Glide module of Schrodinger Suite for extra precision (XP) docking rather than SP docking in the scoring function [48,49]. The structure was validated first by removing the ligand from the active site of M^pro^, and then re-docked with the same ligand to ensure exact binding and less deviation compared to the actual co-crystallized complex. After validation, the final M^pro^ was docked with the selected 20 ligands following the glide XP docking protocol. In case of ligand molecules, van der Waals scaling factors and partial charge cut off were limited to 0.8 and 0.15, respectively. The OPLS2005 force field was used to perform the energy-minimization process. After docking, the top four hits were selected by following the exact alignment with the co-crystal ligand of M^pro^, and were subjected to molecular dynamics simulations [47].

### 4.4. Prime MM-GBSA for Affinity Prediction

To evaluate the actual binding energies of the compounds, the complexes generated from the docking simulation were subjected to MM-GBSA analysis of the prime module [50]. Using the OPLS_AA molecular mechanics force field, MM-GBSA calculates the relative binding energy by combining molecular mechanics energies (EMM), an SGB solvation model for polar solvation (GSGB), and a non-polar solvation term (GNP) composed of a non-polar solvent accessible surface area and van der Waals interactions [51].

The total free energy of binding:ΔG_bind_ = G_complex_ − (G_protein_ + G_ligand_)

### 4.5. Molecular Dynamics Simulation

To better understand protein–ligand complex behavior in an aqueous system, we performed molecular dynamics simulations for selected hits using the Desmond module of Schrödinger Suite 2017-1 (LLC, New York, NY, USA), while the OPLS3 force field was applied. A cubic periodic boundary box surrounding each complex was set up in the presence of a defined solvent. The entire system was set up to equilibrate with the solvated transferable intermolecular potential 3P (TIP3P) water model, which influences and contributes to protein–ligand binding through intermolecular interactions. The boundary conditions were limited to 10° in each direction. To neutralize the entire solvated system, additional counter ions of Na^+^ and Cl^−^ (0.15 M) were added to the system. Then, the entire system was subjected to an energy minimization process, where the Desmond trajectories followed a total of eight stages of a relaxation protocol. Prior to the simulation, the entire system was subjected to an NPT ensemble [52], which maintained a temperature of 300 K [53] and a pressure of 1.01325 bar [54,55]. In stage two, for solute heavy atoms, the Brownian dynamics followed for 12 ps, which followed the NVT ensemble at 10 k. In the third stage, the NVT ensemble was used on the solute heavy atoms for 12 ps maintaining 10 K temperature. Subsequently, NPT was employed with the exception of NVT, where 12 ps simulations at 1 bar pressure were executed in the fourth stage. To solvate the protein cavity, a short simulation of 12 ps was performed in stage five. Before running the final simulation, stages six and seven were employed for a 12 ps simulation under the NPT ensemble, where stage seven was followed without any restraints on solute heavy atoms. The final simulation was run for 700 ns, with a total of 7000 frames maintaining a 100 ps interval between each frame. After the simulation, trajectories were transferred to analyze the RMSD, RMSF, and Rg plots for each complex.

## 5. Conclusions

The report of the current study predicted two possible shikonin derivatives which were screened from 20 shikonin derivatives using a number of computational tools. To gain more structural insight into the M^pro^ binding mechanism and the impact of structural stability on ligand binding, we performed MDS. Comparatively, alpha-methyl-n-butyl shikonin and beta-hydroxyisovaleryl shikonin may inhibit M^pro^ replication. RMSD, RMSF, DCCM, and PCA data were analyzed to predict the most flexible region in the entire structure. In summary, this study claims two feasible lead compounds based on molecular docking and molecular dynamic simulation study, which may provide the possibility of M^pro^ inhibition of SARS-CoV-2. To further support the claim, in vitro and in vivo validations are required to give pace on future research. In addition, a 3D-QSAR study can be conducted in future research to find out the structure activity relationship from the two lead compounds identified.

## Figures and Tables

**Figure 1 ijms-24-03100-f001:**
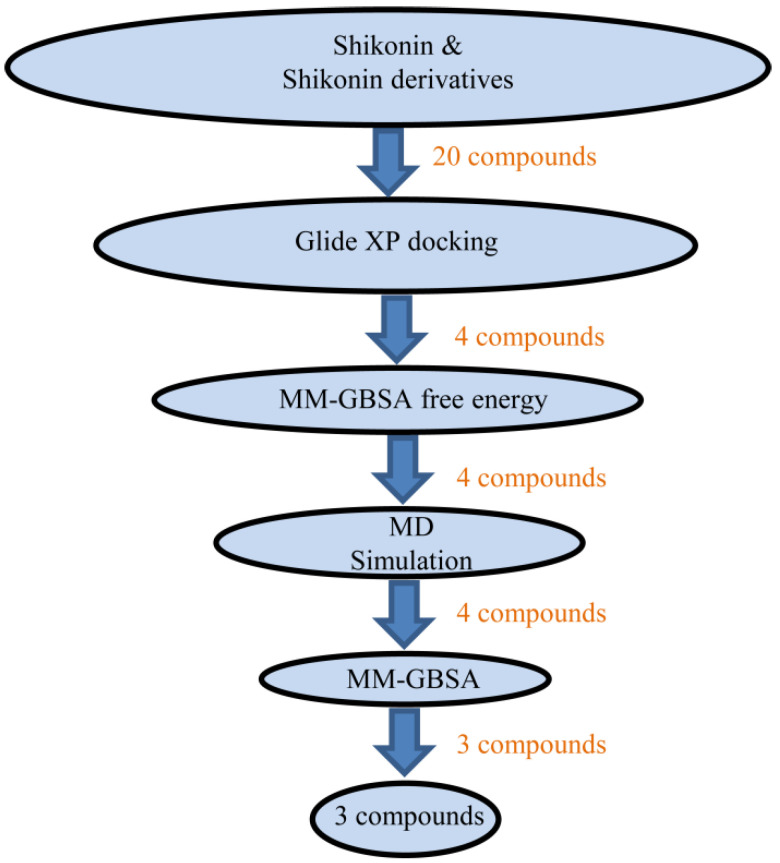
Schematic in silico workflow conducted in the present study.

**Figure 2 ijms-24-03100-f002:**
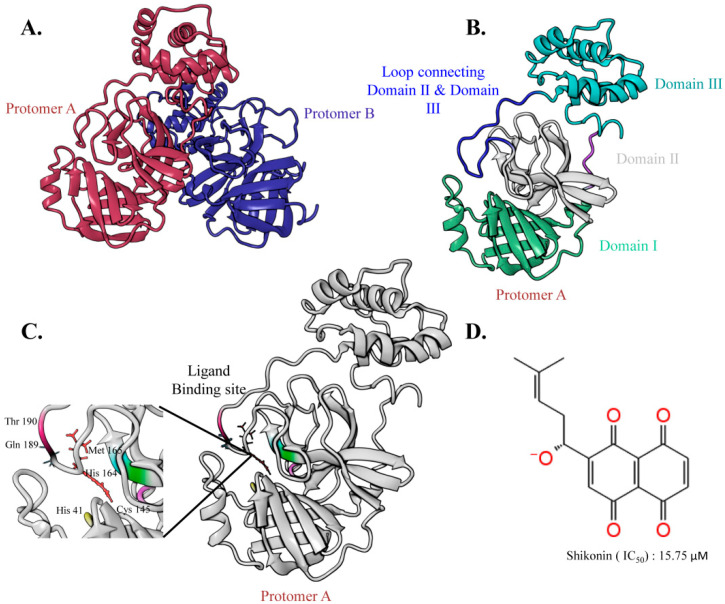
Graphical representation of two protomers of the main protease (M^pro^) of SARS-CoV-2. (**A**) X-ray crystal structure of the M^pro^ homodimer of SARS-CoV-2 (PDB: 7CA8). (**B**) Domain I (residues 10–99), domain II (residues 100–182), domain III (residues 198–303), and the loop connecting domain II & domain III (residues 183–198) of protomer A. (**C**) Ligand-binding site of protomer A (PDB: 7CA8). (**D**) 2D structure of shikonin.

**Figure 3 ijms-24-03100-f003:**
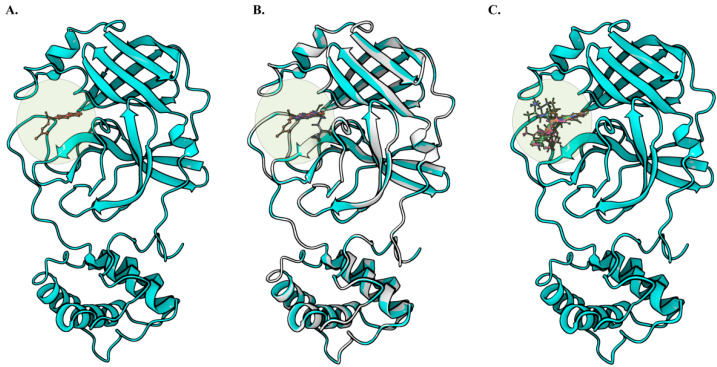
Ligand binding site of the M^pro^ of SARS-CoV-2. X-ray crystal structure of the M^pro^ of protomer (**A**) of SARS-CoV-2 (PDB: 7CA8), and the shikonin binding site, (**B**) the superimposition of docked pose of reference compound with crystal shikonin bound structure, and (**C**) the top four candidate compounds at the same binding site after molecular docking.

**Figure 4 ijms-24-03100-f004:**
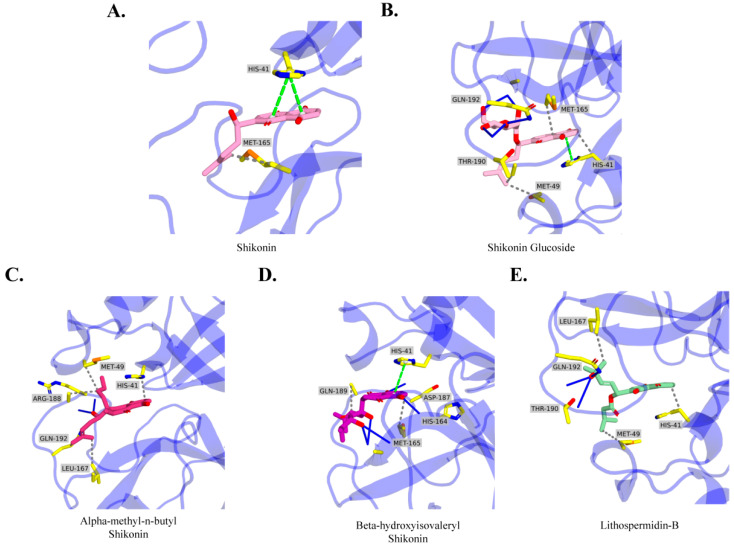
Representation of intermolecular 2D interactions after molecular docking. 2D interaction of (**A**) shikonin, (**B**) shikonin glucoside, (**C**) alpha-methyl-n-butyl shikonin, (**D**) beta-hydroxyisovaleryl shikonin, and (**E**) lithospermidin-B.

**Figure 5 ijms-24-03100-f005:**
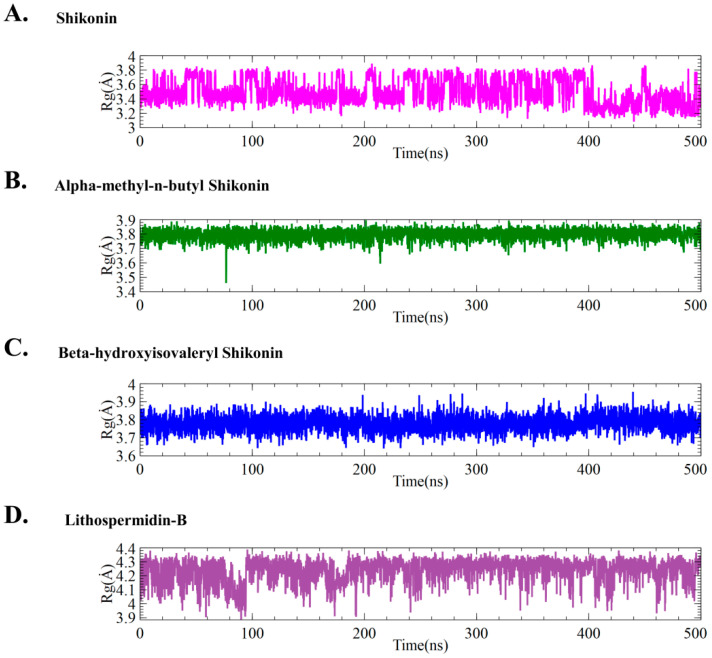
Conformation stability of M^pro^ by means of radius of gyration of backbone atom calculated during the last 500 ns (200–700 ns) of a simulation for both ligand-free and protein–ligand complex systems. Different colors denote different ligands.

**Figure 6 ijms-24-03100-f006:**
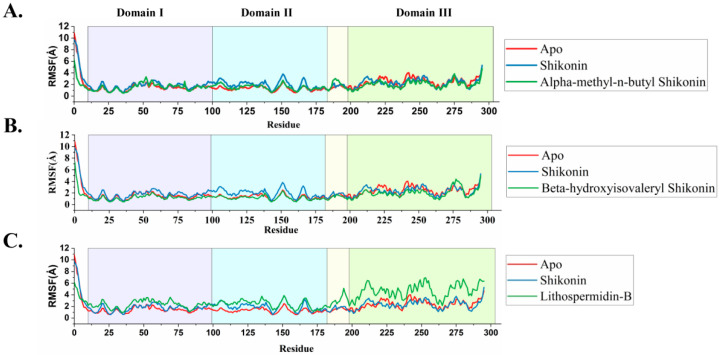
Molecular dynamics simulation study of the selected top hits. Root means square fluctuation (RMSF) of Cα atoms of each docked complex are shown here, representing color for apo (red), shikonin (blue), and the top three hits (alpha-methyl-n-butyl shikonin, beta-hydroxyisovaleryl shikonin, and lithospermidin-B) (green) in (**A**–**C**), respectively.

**Figure 7 ijms-24-03100-f007:**
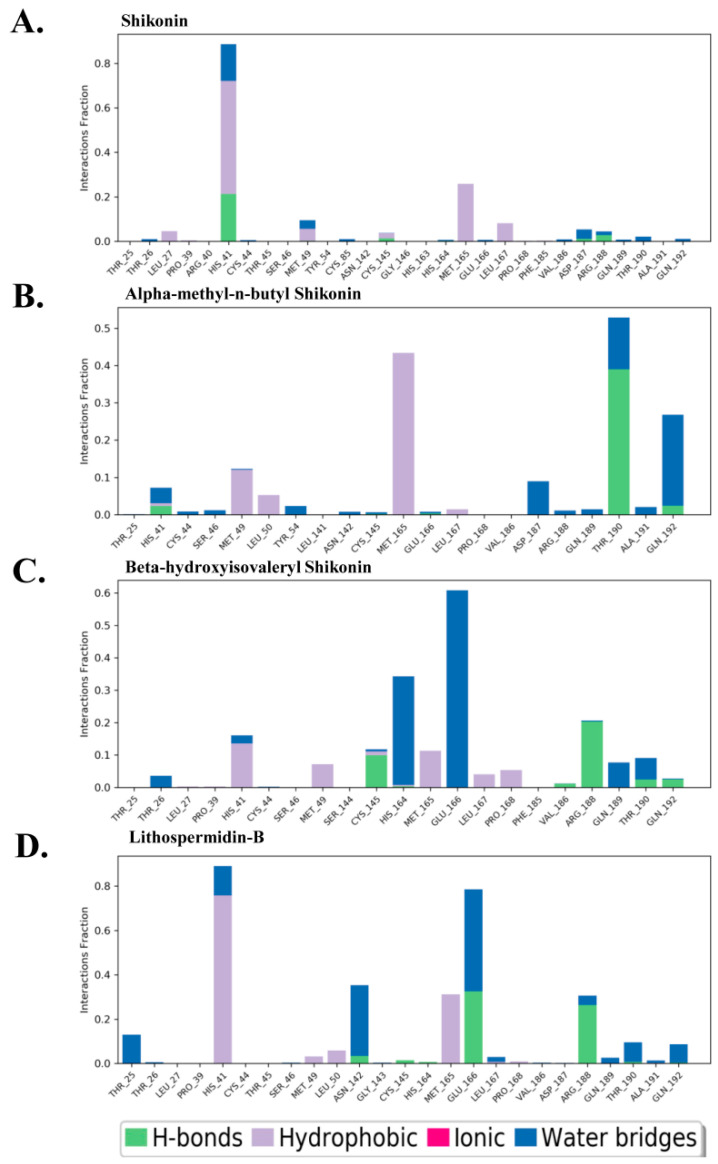
Interaction between proteins and ligands, or the docked complex contact profile. Protein–ligand contacts of following (**A**) shikonin, (**B**) alpha-methyl-n-butyl shikonin, (**C**) beta-hydroxyisovaleryl shikonin, and (**D**) lithospermidin-B. Green, grey, blue, and red indicate hydrogen bonding, hydrophobic interactions, water bridge interactions, and ionic bonding, respectively. The y-axis indicates the total fraction of individual residue interaction in the entire simulation.

**Figure 8 ijms-24-03100-f008:**
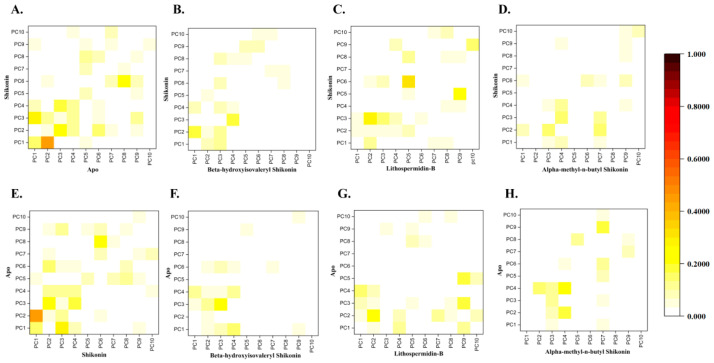
Principle component analysis used to discover variations in the dynamics of the protein. The RMSIP values of the first PC were counted and shown as a gradient heat map from yellow to dark red (indicate low and high values, respectively) to show the similarity and differences between the conformational spaces of wild and hits containing structures. Porcupine plots were employed to show the contributing motions for shikonin alpha-methyl-n-butyl shikonin, beta-hydroxyisovaleryl shikonin, and lithospermidin-B.

**Figure 9 ijms-24-03100-f009:**
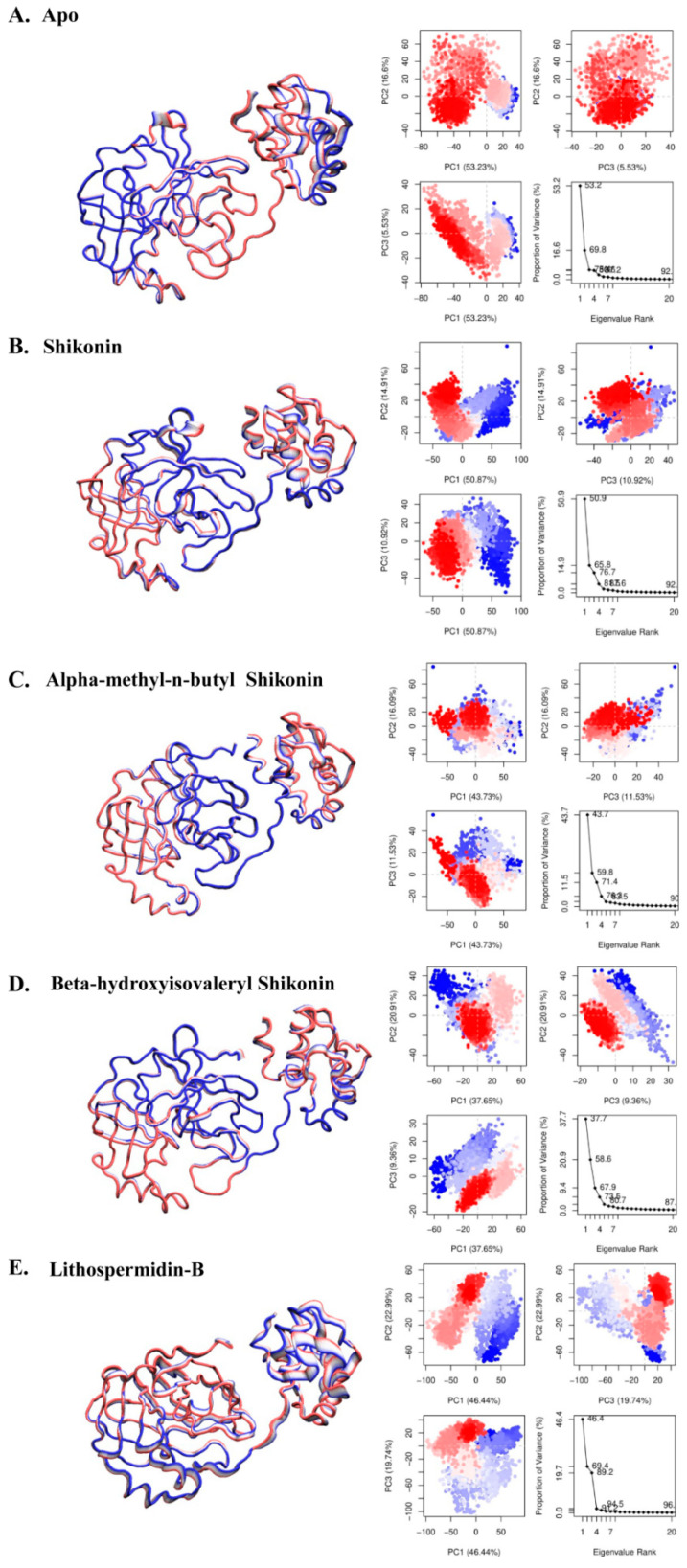
Principle component analysis plotted for (**A**) apo, (**B**) shikonin, (**C**) alpha-methyl-n-butyl shikonin, (**D**) beta-hydroxyisovaleryl shikonin, and (**E**) lithospermidin-B.

**Figure 10 ijms-24-03100-f010:**
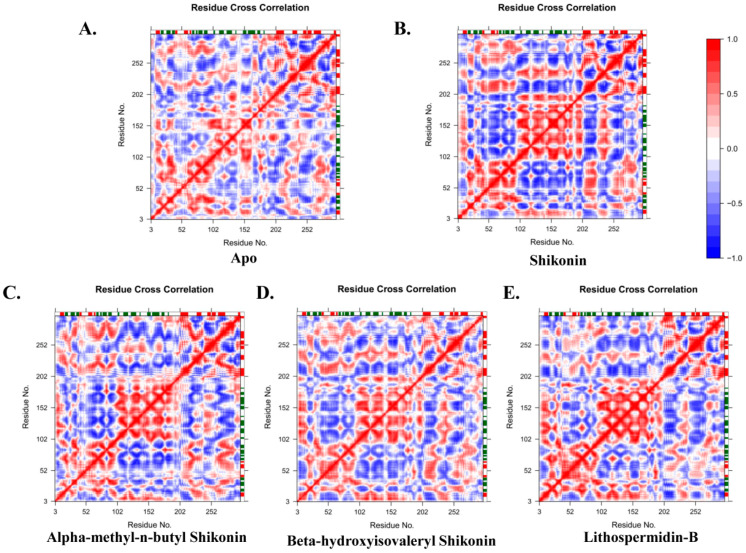
Compound effect on the dynamics of protein movement revealed from simulated trajectories. Graphical representation of correlated dynamic cross-correlation matrix (DCCM) for apo, shikonin, alpha-methyl-n-butyl shikonin, beta-hydroxyisovaleryl shikonin, and lithospermidin-B. The cross-correlations between residues are visualized in a two-dimensional heatmap using a color-coded representation ranging from blue to white to red. Correlated residual movement is indicated by a red color, a blue color indicates negative correlation, and random motions are indicated by white.

## Data Availability

Not applicable.

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
