# Peer review of "Unveiling the Potentiality of Shikonin Derivatives Inhibiting SARS-CoV-2 Main Protease by Molecular Dynamic Simulation Studies"

_ijms, 2023, doi:10.3390/ijms24043100_

Round 1
Reviewer 1 Report
The current MS can not be published in its current form. The below issues should be carefully addressed.
The title should be modified to reflect the studies done in this work, by adding docking and molecular dynamic studies.
English needs careful revision, there are many grammatical and typing mistakes. The proper verb tenses should be used throughout the whole MS. All reference brackets are stuck with the text, please leave space.
The name, alpha–methyl-n-butyl shiko- nin, should be corrected.
The plant family name should be added.
To be easy to follow up on the work results, a figure for all tested derivatives should be added.
For the results to be more significant and reliable a legend other than shikonin should be examined and the obtained results of the derivatives should be discussed in comparison to the results of the standard legend.
Overlay of the docked pose of reference compound with its crystal structure conformation
,, Considering the highest score of 131 binding free energy, four hit compounds were selected.,, What are these four hits?
A table of the hits with the highest docking scores should be included in the MS.
According to the obtained results, authors could suggest structure-activity relationships and what are the core parts in these derivative structures that could be responsible for the activity.
The conclusion should be modified to include the authors' suggestions for future research areas on active hits.
Author Response
Point 1: The title should be modified to reflect the studies done in this work, by adding docking and molecular dynamic simulation.
Response 1: According to the reviewer's suggestion, we corrected the title. The new title is "Unveiling the potentiality of Shikonin derivatives inhibiting SARS-Cov-2 main protease through Molecular Dynamic Simulation studies."
Point 2: English needs careful revision, there are many grammatical and typing mistakes. The proper verb tenses should be used throughout the whole MS. All references brackets are stuck with the text, please leave space.
Response 2 : As reviewer's suggestion, we corrected english grammer and typing mistake through Editage Service. And we correcte make the space between reference bracket and text.
Point 3 : The name alpha-methyl-n-butyl shiko- nin, should be corrected.
Response 3 : The name has been updated to "alpha-methyl-n-butyl shikonin."
Point 4: The plant family name should be added.
Response 4 : The plant family name has been added.
Point 5 : To be easy to follow up on the work results, a figure for all tested derivatives should be added.
Response 5: The authors provided selected derivatives' 2D figures in the supplementary file (Table S1), where Table S1 also contains the original compound name, IUPAC name, docking score, and reference. To avoid the redundancy of same chemical structural figure, the authors didn't place a whole figure of all tested derivatives in the MS file.
Point 6 : For the results to be more significant and reliabe a legend other than shikonin should be examined and the obtained results of the derivatives should be discussed in comparison to the results of the standard legend.
Response 6: First, we thank the reviewer for the valuable suggestion. According to the author's vision, this study mainly focused on the shikonin binding site based on molecular docking and molecular dynamic simulation because the ShiMpro crystal structure highlighted a new binding site (DOI: 10.1016/j.scib.2929.10.018). To validate and get reliable results, the authors extended the simulation period to 700ns. The authors haven't found any published paper on Mpro inhibition incorporating more than 100ns simulation runs. Therefore, the authors agreed to publish the study in the current version, but based on the reviewer's suggestion, the authors will consider the advice in future research.
Point 7: Overlay of the docked pose of reference compound with its crystal structure confirmation.
Response 7: According to the reviewer's suggestion, figure(3B) represents the overlay of the docked pose of the reference compound with its crystal structure confirmation.
Point 8: ,, Considering the highest score of 131 binding free energy, four hits compounds were selected.,, What are these four hits?
Response 8 : At first, the authors didn't mention anywhere in the manuscript, "Considering the highest score of 131 binding free energy, four hits compounds were selected.,,. In section 4.3, it is clearly written that the top four hits were determined by following the exact alignment with the co-crystal ligand of Mpro. In section 2.2, the authors also mentioned that the highest binding energy is considered for selecting top four hits. To clear confusion among the readers, the authors updated the writing in the result portion. The score of binding free energy is included in the supplementary file ( Table S3).
Point 9 : A table of the hits with highest docking scores should be included in the MS.
Response 9: The selected hit's highest docking scores have already been included in the supplementary file ( Table S3).
Point 10: According to the obtained results, authors could suggest structure-activity relationships and what are the core parts in these derivatives structures that could be responsible for the activity.
Response 10: At first, all authors are thankful to the reviewer's significant comment and appreciate the suggestion of conducting structure-activity relationships. According to the author's vision, a 3D-QSAR study will be considered in future research.
Point 11: The conclusion should be modified to include the authors' suggestion for future research areas on active hits.
Response 11: The conclusion has been modified based on the author's suggestions.
We corrected the spelling and removed extra spaces in the final MS file. With your kind consideration, we have attached the updated MS file.
Reviewer 2 Report
The paper reads nicely and shows a good use of language. Also, the citations and links to tools and software are well covered. The authors have presented their results in a transparent way and have a good storyline in the paper. The conclusion is nicely inferred from the results presented.
My only comment is that the authors check in pubchem or other databases for the existence of the newly designed or proposed molecules (Shikonin derivatives) with MPro activity and report them and their eventual use.
In future research, the authors should consider conducting a 3D-QSAR study with the structure of the most active compounds identified in this study, in order to understand their structure-activity relationship and propose new bioactive compounds.
Author Response
Point 1: The paper reads and shows a good use of language. Also, the citations and links to tools and software are well covered. The authors have presented their results in a transparent way and have a good storyline in the paper. The conclusion is nicely inferred from the results presented.
Response 1: All authors are thankful to the reviewer for spending valuable time for reviewing the work, as well as appreciating the work.
Point 2: My only comment is that the authors check in pubchem or other databases for the exixtence of the newly designed or proposed molecules ( Shikonin derivatives ) with Mpro activity and report them and their eventual use.
Response 2: All authors thank the reviewer for the valuable suggestion of searching already reported shikonin derivatives, which have activity against Mpro of SARS CoV-2. After the literature review, authors found two shikonin derivatives (Beta, beta-dimethylallyl shikonin, and Acetylshikonin) that showed activity against Mpro inhibition in the previous report; however, we already mentioned the docking score, and related reference paper of those compounds in supplementary file ( Table S1), as well reference papers were cited in main MS file. However, due to the author’s vision and storyline, only four derivatives were selected from the listed derivatives in the primary stage, which showed an almost identical docked position with a crystal shikonin structure.
Point 3 : In future research, the authors should consider conducting a 3D-QSAR study with the structure of the most active compounds identified in this study, in order to understand their structure-activity relationship and propose new bioactive compounds.
Response 3 : All authors highly appreciate the suggestion and consideration of 3D-QSAR study in future research. Based on our plan and vision, we will conduct the 3D-QSAR study to understand the structure-activity relationship of proposed compounds in our future research.
We corrected the spelling and removed extra spaces in the final MS file. With your kind consideration, we have attached the updated MS file.
Reviewer 3 Report
In this manuscript, Das and co-authors unveiled the potentiality of Shikonin derivatives inhibiting 2 SARS-CoV-2 main protease. The authors were able to screen 20 shikonin derivatives from which few derivatives showed higher binding affinity than shikonin. In this study, they found few shikonin derivatives may have the capability to inhibit the M-pro replication process which could help researchers to find shikonin derivatives that inhibit the M-pro process in SARS-CoV-2.
Besides this, the manuscript is well-written, and the content is explained scientifically. Based on the potential and usefulness of the present manuscript, I would recommend it for publication in the International Journal of Molecular Sciences.
Author Response
Point 1: In this manuscript, Das and co-authors unveiled the potentiality of Shikonin derivatives inhibting 2 SARS-CoV-2 main protease. The authors were able to screen 20 shikonin derivatives from which few derivatives showed higher binding affinity than shikonin. In this study, they found few shikonin derivatives may have the capability to inhibit the M-pro replication process which could help researchers to find shikonin derivatives that inhibit the M-pro process in SARS-CoV-2.
Response 1: All authors are thankfull to the reviewer for the appreciation of this work.
Point 2: Besides this, the manuscript is well-written and the content is explained scientifically. Based on the potential and usefulness of the present manuscript, I would recommend it for the publication in the International Journal of Molecular Sciences.
Response 2: Again, all authors are thankfull to the reviewer for recommending to publish this work in the International Journal of Molecular Sciences.
We corrected the spelling and removed extra spaces in the final MS file. With your kind consideration, we have attached the updated MS file.
Round 2
Reviewer 1 Report
It is not clear, where the corrections have been done in the MS. Please, highlight the corrections in your revised MS To be easy to revise.
Round 3
Reviewer 1 Report
No comment